# Research on Sinomenine Inhibiting the cGAS-STING Signaling Pathway to Alleviate Renal Inflammatory Injury in db/db Mice

**DOI:** 10.3390/ph18070934

**Published:** 2025-06-20

**Authors:** Xiaofei Jin, Tongtong He, Tianci Zhang, Xiaorong Wang, Xiangmei Chen, Bin Cong, Weijuan Gao

**Affiliations:** 1Hebei Key Laboratory of Chinese Medicine Research on Cardio-Cerebrovascular Disease, Hebei University of Chinese Medicine, No. 3 Xingyuan Road, Luquan District, Shijiazhuang 050200, China; jxf1655@163.com (X.J.); halona0211@163.com (T.H.); zyky2020@126.com (T.Z.); 18479700567@163.com (X.W.); 2National Clinical Research Center for Kidney Diseases, Department of Nephrology, First Medical Center of Chinese PLA General Hospital, Beijing 100853, China; xmchen301@126.com; 3Hebei Key Laboratory of Forensic Medicine, College of Forensic Medicine, Hebei Medical University, Shijiazhuang 050017, China

**Keywords:** sinomenine, diabetic nephropathy, inflammatory injury, cGAS, STING

## Abstract

**Objectives:** This study aims to elucidate the potential molecular mechanism of Sinomenine (SIN) in treating renal injury in Diabetic Nephropathy (DN) through network pharmacology, molecular docking, and in vivo validation. **Materials and Methods**: db/db mice were used as a DN model to evaluate the therapeutic effects of SIN on body weight, blood glucose levels, renal function, and histopathology. Network pharmacology and molecular docking were integrated to predict the potential molecular mechanisms of SIN in DN treatment. Subsequently, in vivo validation was performed on db/db mice using ELISA, Western blotting, RT-qPCR, immunofluorescence, and immunohistochemistry. **Results**: Firstly, we found that SIN (62.4 mg/kg) improved general conditions and renal function in db/db mice, alleviating renal pathological damage. Network pharmacology analysis identified IL-1β, IL-6, and TNF-α as key targets of SIN in DN. SIN reduced IL-1β, IL-6, and TNF-α levels by inhibiting the cGAS/STING signaling pathway and its downstream p-TBK1, p-IRF3, and NF-κB expression. **Conclusions**: SIN alleviates inflammatory injury in DN, potentially through the cGAS/STING pathway.

## 1. Introduction

Diabetic nephropathy (DN) is a serious microvascular complication of diabetes mellitus (DM), with a rising prevalence that places a growing burden on global healthcare systems and affected individuals [1]. As DN advances, the glomerular filtration rate initially increases, leading to clinical manifestations such as proteinuria and edema [2] and ultimately progressing to end-stage renal disease (ESRD) [3,4]. The underlying mechanisms of DN are highly intricate, encompassing hyperglycemia-induced inflammation, oxidative stress, the accumulation of advanced glycation end products (AGEs), and microvascular dysfunction, among other factors [5,6]. In recent years, immune system dysregulation and chronic inflammation have been increasingly recognized as pivotal contributors to DN pathogenesis [7]. Persistent inflammation throughout DN progression, driven by prolonged hyperglycemia, promotes the release of pro-inflammatory cytokines and activates signaling pathways such as NF-κB [8], ultimately exacerbating chronic renal inflammation and accelerating disease progression [9].

A key driver of inflammatory injury in DN is the activation of the cGAS/STING signaling pathway [10]. Cyclic GMP-AMP synthase (cGAS) acts as a cytosolic DNA sensor, detecting both self-DNA from cellular damage and foreign DNA, subsequently catalyzing the production of cyclic GMP-AMP (cGAMP), which in turn activates the stimulator of interferon genes (STING) [11]. Upon activation, STING triggers a cascade of downstream signaling events, including the NF-κB pathway and type I interferon responses, leading to immune cell recruitment and the secretion of pro-inflammatory cytokines, thereby intensifying renal inflammation [12]. In DN models, dysregulated activation of the cGAS/STING pathway has been closely linked to renal inflammation, fibrosis, and progressive renal dysfunction [13]. Consequently, targeting this pathway for inhibition may represent a promising therapeutic approach to mitigate DN progression.

Qing Feng Teng (scientific name: *Sinomenium acutum* (Thunb.) Rehd. et Wils) is a traditional Chinese medicinal herb known for its ability to promote blood circulation and facilitate urination. Clinically, it has been utilized in the treatment of diabetic nephropathy (DN), chronic nephritis, and other conditions [14]. Sinomenine (SIN), the principal bioactive alkaloid of *Sinomenium acutum*, exhibits potent anti-inflammatory and immunomodulatory effects [15,16]. Previous studies have highlighted the nephroprotective and cardiovascular benefits of SIN, as well as its anti-cancer properties, efficacy in rheumatoid arthritis treatment, and analgesic potential [17,18]. Furthermore, SIN has been demonstrated to reduce renal inflammation and alleviate DN-induced kidney damage by modulating the JAK2/STAT3/SOCS1 pathway [19]. It has been reported that SIN can significantly improve glomerular endothelial cell dysfunction induced by high glucose. Additionally, SIN has been found to mitigate renal fibrosis by influencing miR-204-5p expression and regulating the PI3K-AKT pathway [20]. However, whether SIN alleviates renal inflammatory injury in DN through modulation of the cGAS-STING signaling pathway remains largely unexplored.

In this study, db/db mice were utilized to establish an in vivo model of DN. Initially, the impact of SIN on blood glucose levels, biochemical markers, and renal pathology was assessed to determine its therapeutic efficacy in DN. Subsequently, leveraging network pharmacology predictions, in vivo experiments were performed to further elucidate the nephroprotective mechanisms of SIN, primarily through the suppression of the cGAS/STING signaling pathway and its associated inflammatory response in db/db mice.

## 2. Results

### 2.1. SIN Effectively Prevents the Progression of Renal Dysfunction in db/db Mice

The db/m mice exhibited normal growth conditions, displaying active behavior, quick responses, glossy fur, stable food and water intake, and light-yellow urine. In contrast, as time progressed, db/db mice exhibited signs of lethargy or reduced activity, clumsy movements, obesity, dry and dull fur, and classic symptoms of polyuria, polydipsia, and polyphagia (Figure 1A). During the experiment, body weight and blood glucose levels continued to rise in the MOD group. Although these parameters also increased in the SIN-treated groups, SIN administration effectively slowed their rate of increase (Figure 1B,C). As shown in Figure 1C, SIN and CaD significantly reduced serum HbA1c levels. Furthermore, renal function assessments revealed that SIN and CaD treatment reduced 24 h mALB, Cys-C, Scr, and BUN, indicating its protective effects on kidney function (Figure 1E–H).

### 2.2. SIN Improves Renal Pathological Damage in db/db Mice

The pathological structure was scored according to the Chronic Kidney Disease Pathological Injury Scoring System (CKD-PIS), and the results are shown in Table 1. As shown in Figure 2, HE staining revealed that the CON group mice exhibited normal renal tubules, glomeruli, and interstitial structures. In contrast, the MOD group showed severe renal structural abnormalities, including tubular dilation, epithelial necrosis, and interstitial expansion. SIN and CaD administration effectively alleviated these pathological changes in db/db mice. Among the different dosage groups, the SIN-M group showed the most significant improvement in renal pathology, whereas the SIN-L and SIN-H groups exhibited moderate amelioration but not to the extent observed in the SIN-M group. PAS staining further confirmed that the glomerular region of MOD was stained deep red, with a significant increase in glycogen granules in the renal tubules. In some renal tubule epithelial cells, glycogen granules showed abnormal aggregation, forming clumps of purplish-red areas. In contrast, the CON group exhibited a uniform light red color without obvious glycogen deposition. SIN and CaDtreatment can reduce glycogen deposition in both the glomeruli and renal tubules, with the most significant improvement observed in the SIN-M group. Interestingly, TEM analysis showed that in db/db mice, mitochondria were swollen and deformed, cristae disappeared, foot cells fused widely, and the basement membrane thickened significantly. In the SIN-M group, mitochondrial damage was improved, foot fusion was significantly reduced, and basement membrane thickening was not obvious (Figure 2), indicating that SIN could effectively reduce DN-related renal tissue ultrastructural damage.

### 2.3. Network Pharmacology Analysis of SIN and DN

A comprehensive network pharmacology analysis was conducted to elucidate the potential molecular targets of SIN in the therapeutic management of DN. Initially, 258 SIN-specific targets were retrieved from the TCMSP and PharmMapper databases, whereas 1348 DN-associated targets were identified through searches in Genecards, OMIM, and TTD. The intersection analysis of SIN and DN-related targets pinpointed 57 overlapping targets, which are posited as the prospective therapeutic targets of SIN for DN intervention (Figure 3A,B). These 57 overlapping targets were subsequently input into the STRING database to construct a protein–protein interaction (PPI) network (Figure 3C). Visualization of the network was executed using Cytoscape (version 3.9.1) software (Figure 3D), revealing a network structure comprising 51 nodes and 441 edges. The CentiScaPe plugin within Cytoscape was employed to discern core genes. The identified core genes-IL-1β, TNF-α, and IL-6-exhibited elevated degrees and more extensive connectivity to adjacent targets in comparison to other target genes.

### 2.4. KEGG and GO Analysis

To elucidate the underlying mechanisms of SIN in the treatment of DN, an enrichment analysis of the identified targets was conducted utilizing the DAVID database. A bioinformatics platform was employed to depict the distribution of therapeutic targets across various hierarchical levels (Figure 4). The Gene Ontology (GO) analysis pinpointed the top 10 enriched categories within biological processes (BP), cellular components (CC), and molecular functions (MF), with prioritization based on the number of associated genes (Figure 4A). Concurrently, the KEGG analysis delineated the top 12 most pertinent signaling pathways, as illustrated in Figure 4B. Our findings demonstrate that SIN targets in DN therapy are predominantly enriched within biological processes and molecular functions pertaining to the regulation of inflammation, cytokine receptor interaction, and allied mechanistic pathways. Furthermore, these targets are specifically localized within the cytoplasmic compartment, including the endoplasmic reticulum membrane and the outer aspect of the plasma membrane. The therapeutic efficacy of SIN in the context of DN is mediated through pivotal signaling pathways, such as the tumor necrosis factor (TNF) signaling pathway, the cytosolic DNA-sensing pathway, and pathways associated with lipid metabolism and atherosclerosis, among other contributory pathways.

### 2.5. Molecular Docking of SIN with Core Targets

To elucidate the mechanistic underpinnings of SIN therapeutic efficacy in treating DN and its potential binding interactions with pivotal DN-associated targets, molecular docking analyses were performed between SIN and these essential targets (Figure 5). Generally, lower binding energy correlates with a more stable ligand–receptor interaction, suggesting a stronger binding affinity of the compound to the protein. Our molecular docking results indicated that SIN could spontaneously interact with potential key DN targets, including TNF-α, IL-1β, and IL-6, as well as critical proteins within the signaling pathway, such as cGAS, STING, and NF-κB. Notably, the binding energy observed with STING, a pivotal protein in the pathway, was the lowest among the tested interactions. It is pertinent to mention that TNF-α, IL-1β, and IL-6 are downstream effector molecules within the cGAS-STING signaling pathway. These findings suggest that the anti-inflammatory properties of SIN may be mediated through the modulation of the cGAS-STING pathway.

### 2.6. SIN Inhibits Inflammatory Injury in db/db Mice

To validate the targets predicted through network pharmacology and confirm their involvement in DN pathogenesis, we evaluated key inflammatory markers. Immunofluorescence staining revealed a significant increase in the fluorescence intensity of NF-κB and IL-1β in the MOD group compared to the CON group, indicating heightened renal inflammation. However, SIN treatment markedly reduced the fluorescence intensity of these markers, suggesting an anti-inflammatory effect. Notably, the PDTC group displayed fluorescence intensities similar to those observed in the SIN-treated group (Figure 6A–D). Meanwhile, the DMSO control group exhibited fluorescence intensity levels comparable to those of the MOD group, confirming that the solvent itself had no significant impact.

Further analysis of pro-inflammatory cytokine levels showed a substantial elevation of IL-1β, IL-6, TNF-α, and MCP-1 in the serum of MOD group mice compared to the CON group (Figure 6E–H). SIN treatment effectively lowered the serum concentrations of these cytokines, demonstrating an effect comparable to PDTC. Similarly, in renal tissues, the expression levels of IL-1β, IL-6, and TNF-α were highest in the MOD group, whereas both SIN and PDTC treatments significantly suppressed their production (Figure 6I–K). These findings further support the anti-inflammatory role of SIN in DN through the inhibition of key inflammatory mediators.

### 2.7. SIN Alleviates Renal Inflammatory Injury in db/db Mice by the cGAS/STING Signaling Pathway

To further investigate the molecular mechanism by which SIN alleviates inflammatory injury via the cGAS/STING signaling pathway, we analyzed key markers associated with this pathway. TBK1 and IRF3, as critical downstream targets of cGAS/STING activation, were examined along with cGAS and STING using immunohistochemical (IHC) staining. As illustrated in Figure 7A–E, IHC analysis of renal tissues demonstrated a significant increase in cGAS, STING, TBK1, and IRF3 expression in the MOD group compared to the CON group. SIN treatment markedly attenuated this upregulation. Notably, the administration of C-176, a well-established STING inhibitor, resulted in a comparable suppression of cGAS and STING expression, indicating that SIN exerts its effects through modulation of this pathway. Meanwhile, the DMSO control group exhibited no significant changes compared to the MOD group, confirming that the solvent itself had no therapeutic effect.

Western blotting analysis (Figure 8A–E) further substantiated these findings, revealing a substantial upregulation of key cGAS/STING pathway proteins in the MOD group, along with significantly increased phosphorylation of TBK1 and IRF3. However, SIN treatment effectively downregulated NF-κB, cGAS, and STING expression while also reducing the phosphorylation levels of TBK1 and IRF3, consistent with the IHC staining results.

Additionally, RT-qPCR analysis was performed to assess the mRNA expression of cGAS, STING, TBK1, and IRF3 in renal tissues. As anticipated, SIN treatment significantly decreased the mRNA levels of these key genes compared to the MOD group, further confirming its role in inhibiting cGAS/STING-mediated inflammation.

## 3. Discussion

The incidence and mortality of DN are both high, and it has been shown to be associated with hyperglycemia and inflammatory responses [21]. Conventional treatments have limited efficacy in maintaining renal function; hence, we found great significance in exploring the effective treatment of DN. Medicinal plants and their active components play a crucial role in the treatment of DN. SIN, as the main active component of Qing Feng Teng, has been proven to possess various pharmacological effects [22,23]. Due to its excellent clinical outcomes and minimal side effects, it demonstrates unique pharmacological mechanisms in renal diseases [24]. Studies have shown that in chronic kidney disease mouse models, SIN can inhibit inflammatory responses and protect kidney function [25]. Research by Li et al. suggests that SIN can reduce the infiltration of inflammatory cells, thereby protecting against IgA nephropathy [26]. SIN has also shown significant efficacy in the treatment of interstitial nephritis [27]. In this study, we used network pharmacology to predict the targets of SIN in treating DN, validated the therapeutic effect of SIN through molecular docking and animal experiments, and found that SIN exerts a renoprotective effect by inhibiting the production of NF-κB, which reduces the secretion of inflammatory factors. Specifically, SIN effectively lowers the expression of NF-κB by inhibiting the cGAS/STING signaling pathway, thereby ameliorating inflammatory damage in DN. This provides new insights into the treatment of DN.

The db/db mouse model originates from the C57BL/KSJ inbred strain through autosomal recessive inheritance. A leptin gene mutation leads to obesity, hyperglycemia, and insulin resistance, making it one of the best models for DN research [28]. Studies have confirmed that db/db mice exhibit renal dysfunction at 12 weeks of age [29]. In this study, 12-week db/db mice were used as a DN animal model, and SIN intervention was administered. The results showed that SIN could suppress blood glucose levels and weight gain in db/db mice while protecting renal function. Further histological analysis of kidney tissues demonstrated that SIN improved structural abnormalities, including glomerular expansion, glycogen deposition, and basement membrane thickening.

In recent years, network pharmacology has played a crucial role in exploring pharmacological mechanisms [30]. In this study, we used network pharmacology to screen the intersecting targets of SIN and DN, identifying a total of 57 targets. The most critical of these are IL-1β, IL-6, and TNF-α, suggesting that the main mechanism by which SIN protects the kidneys may involve the inhibition of pro-inflammatory factor secretion. KEGG and GO analyses further support the multi-target action of SIN in inhibiting inflammation. This multi-target, multi-pathway characteristic suggests that SIN may alleviate the pathological progression of DN by comprehensively regulating upstream DNA sensing mechanisms and downstream pro-inflammatory factors.

Inflammatory damage plays a significant role in DN [31], as local renal inflammation triggers intrinsic renal cell damage and apoptosis [32]. The activated innate immune cells then enhance the autocrine and paracrine pro-inflammatory responses, amplifying the inflammatory reaction and leading to renal dysfunction [33]. NF-κB is an essential nuclear transcription factor in the inflammatory response of DN. Upon activation, it enters the nucleus to initiate the transcription of pro-inflammatory factors [34]. As a result, various pro-inflammatory factors, including IL-1β, IL-6, and TNF-α, increase [35]. NF-κB activates the NLRP3 inflammasome by stimulating the expression of NLRP3 and pro-IL-1β, which is then converted into IL-1β by caspase-1 to mediate inflammation [36,37]. It can also directly bind to the IL-6 gene promoter region, increasing its transcriptional activity and enhancing its secretion [38]. IL-6 can indirectly enhance NF-κB activity through the classic JAK/STAT3 signaling pathway, forming a positive feedback loop [39]. NF-κB dissociates its inhibitory complex and translocates to the nucleus to activate TNF-α transcription, promoting its expression [40]. In vivo experiments showed that SIN significantly reduced the levels of IL-1β, IL-6, and TNF-α in the serum and kidney tissues of db/db mice. Furthermore, immunofluorescence experiments demonstrated that compared to the CON group, the fluorescence intensity of NF-κB and IL-1β in the kidney tissues of the MOD group was significantly enhanced. After SIN treatment, this intensity was notably reduced, and the effect was comparable to that of PDTC, indicating SIN’s inhibitory effect on the inflammatory signaling pathway.

There are many pathways through which inflammation occurs in DN, and the recently emerging cGAS/STING pathway has drawn additional attention. cGAS is an innate immune receptor for cytoplasmic double-stranded DNA (dsDNA), and its main function is to recognize dsDNA in the cytoplasm [41]. In DN, sustained hyperglycemia causes the massive release of intracellular dsDNA [42]. After cGAS binds to dsDNA, it produces cGAMP, which is the first step in activating the cGAS/STING signaling pathway. cGAMP, as a second messenger, binds to and activates STING, which is located in the endoplasmic reticulum [43]. After activation, STING migrates from the endoplasmic reticulum to the ER-Golgi intermediate compartment and recruits TANK-binding kinase 1 (TBK1), triggering a downstream signaling cascade [44]. During this process, the STING-TBK1 signaling complex mediates the polyubiquitination and degradation of NF-κB via the ubiquitin-proteasome pathway [42], which releases NF-κB into the nucleus to promote the transcription of inflammatory cytokines, induce inflammatory cell infiltration, and exacerbate inflammatory damage [45]. Additionally, the cGAS/STING pathway also promotes the phosphorylation of the downstream protein IRF3 [46,47], and these transcription factors, once in the nucleus, further enhance the transcription of inflammatory factors such as TNF-α, IL-6, and IL-1β [48], worsening kidney damage in DN and ultimately leading to renal failure. The experimental results in this study show that SIN can significantly downregulate the expression levels of cGAS, STING, p-TBK1, p-IRF3, and NF-κB in the kidney tissues of db/db mice. Moreover, the inhibitory effect of SIN on STING is comparable to that of C-176. In conclusion, these experimental findings suggest that inhibition of the cGAS/STING signaling pathway could be a potential strategy for SIN in the treatment of DN.

Although this study established the concentration gradient of SIN (sinomenine) and explored its therapeutic effect on db/db mice, there are still multiple limitations. The experiment only verified the short-term therapeutic effect of SIN in the animal model, and it is only at the animal’s overall level. Its mechanism of action at the cellular level and its actual therapeutic effect in humans still need further exploration. In terms of pharmacokinetics, this study did not determine the pharmacokinetic parameters and bioavailability of SIN in db/db mice, and it is impossible to clarify its dynamic changes and utilization efficiency in the body. In the future, our research group will conduct in-depth studies on this.

## 4. Materials and Methods

### 4.1. Reagents

SIN (purity: >98%; Cat#: YK-221102) was provided by Hunan Zhengqing Pharmaceutical Group Co., Ltd. (Huaihua, Hunan, China). CaD (Calcium Dobesilate, Cat#: H20123194) was purchased from Shanghai Haihong Industry (Group) Chaohu Jinchen Pharmaceutical Co., Ltd. (Shanghai, China). The NF-κB protein inhibitor (PDTC, Cat#: HY-18738) was purchased from MedChemExpress (Monmouth Junction, NJ, USA). The STING protein inhibitor C-176 (Cat#: S6575) was obtained from Selleck Chemicals (Houston, TX, USA). DMSO (Cat #: D8418) from Sigma (Burlington, MA, USA). Anti-NF-κB antibody (Cat#: GB11997) and anti-IL-1β antibody (Cat#: GB300002) were both purchased from Sevier Biotechnology Co., Ltd. (Wuhan, Hubei, China). cGAS (Cat#: NM_173386) was obtained from Leading Biology (Richmond, CA, USA), STING (Cat#: 13647), p-TBK1(Cat#: 5483) and p-IRF3(Cat #:79945) purchased from CST (Danvers, MA, USA), and TBK1 (Cat#: sc-73115) and IRF3 (Cat#: sc-33642) were purchased from Santa Cruz Biotechnology (Santa Cruz, CA, USA). cDNA was synthesized using a reverse transcription kit (Cat#: RR047A) purchased from Takara (Kusatsu, Japan). ELISA kits and related reagents were purchased from Shanghai Zhuocai (Shanghai, China).

### 4.2. Animals and Experimental Design

12wks SPF-grade male C57BLKSJ-db/db mice, weighing 40 ± 6 g, and wild-type db/m mice, weighing 22 ± 4 g, both possessing SPF-grade genetic backgrounds, were procured from GemPharmatech Biotechnology Co. (Nanjing, China). Animals employed in the investigation were procured under the designated license number SCXL (Jing) 2024-0009. All animals utilized in this study were granted approval by the Animal Ethics Committee of Hebei University of Chinese Medicine (Approval Number: DWLL202302022). These animals were housed in an environment featuring a 12-h light–dark cycle, with temperature control set at 24 ± 2 °C. Additionally, they received consistent provisions for feeding, hydration, and regular bedding alterations.

#### 4.2.1. Experiment 1

The investigation encompassed six distinct cohorts, 60 subjects were included: 10 db/m mice as the standard control group (CON), five groups of db/db mice randomly assigned to the model group (MOD), and low dose (SIN-L), medium dose (SIN-M) and high dose (SIN-H) SIN treatment group, Calcium Dobesilate (CaD) group. SIN was administered orally by gavage at concentrations of 31.2, 62.4, and 124.8 mg/(kg·day), and CaD was administered at a dose of 195 mg/(kg·day) for 8 weeks. Throughout the 8-week experimental phase, body weight and fasting blood glucose concentrations were systematically assessed at two-week intervals. Subsequent to the treatment period, the animals were euthanized humanely, and blood, 24-h urine samples, and renal tissues were harvested to investigate the impact of SIN on renal dysfunction in db/db mice.

#### 4.2.2. Experiment 2

The investigation encompassed five distinct cohorts, each comprising 10 subjects: a cohort of 10 db/m mice serving as the normative control group (CON) and four cohorts of db/db mice, which were randomly allocated to the model group (MOD), the SIN group (SIN), the PDTC (The NF-κB protein inhibitor) group, and the DMSO group. The PDTC cohort received an oral administration of 30 mg/(kg·day) via gavage, whereas the DMSO control group was administered an equivalent volume of DMSO. Following an 8-week treatment regimen, the mice were humanely euthanized to facilitate comprehensive analysis. Subsequently, blood samples and renal tissue specimens were procured to elucidate the inhibitory impact of SIN on inflammatory-mediated damage in db/db murine models.

#### 4.2.3. Experiment 3

The investigation encompassed five distinct cohorts, each comprising 10 subjects: a cohort of 10 db/m mice serving as the normative control group (CON) and four cohorts of db/db mice, which were randomly allocated to the model group (MOD), the SIN group (SIN), the C-176 (The STING protein inhibitor) group, and the DMSO group. The C-176 group was subjected to daily intraperitoneal administrations of 5 mg/(kg·day), whereas the DMSO control group received an equivalent volume of dimethyl sulfoxide. The administration of both therapeutic regimens persisted over an 8-week duration. Following an 8-week experimental period, the murine subjects were euthanized, and renal tissue samples were excised for subsequent analysis. The objective of this segment of the investigation was to ascertain the potential of SIN to mitigate inflammatory damage via modulation of the cGAS/STING signaling pathway.

### 4.3. Biochemical Assays

Blood urea nitrogen (BUN) and serum creatinine (Scr) concentrations were quantified utilizing the Chemray 800 automated biochemical analyzer (Leidu, Shenzhen, China).

### 4.4. ELISA

According to the instructions of the enzyme-linked immunosorbent assay (ELISA) kit, the following measurements were performed: 24-h urinary mALB; serum levels of HbA1c, Cys-C, MCP-1, IL-1β, IL-6, and TNF-α; and renal tissue levels of inflammatory factors IL-1β, IL-6, and TNF-α.

### 4.5. H&E and PAS

Renal tissue specimens were preserved via fixation in a 10% formalin solution and subsequently embedded in paraffin wax. Serial sections of 4 μm thickness were prepared, deparaffinized using xylene, and subjected to hematoxylin and eosin (HE) and periodic acid-Schiff (PAS) staining protocols. These stained sections were then examined under a light microscope (DM5000B, Leica, Wetzlar, Germany) for histopathological analysis.

### 4.6. TEM

Renal cortical tissue specimens from each experimental cohort were preserved in 2.5% glutaraldehyde at 4 °C. These samples were subsequently sectioned into 1 mm^3^ segments and processed for transmission electron microscopy (TEM) utilizing a JEM-1400 electron microscope (Tokyo, Japan). This procedure was undertaken to examine the ultrastructural features of the kidneys and to acquire high-resolution imagery for further analysis.

### 4.7. Network Pharmacology Analysis

Preliminary experimental data indicated that SIN exerted a protective effect on renal function in db/db murine models. A combination of network pharmacology and molecular docking methodologies was utilized to elucidate the potential underlying mechanisms. Initially, we conducted a comprehensive search for SIN-associated targets utilizing established databases, including SwissTargetPrediction (http://www.swisstargetprediction.ch, accessed on 1 January 2025), TCMSP (https://www.tcmsp-e.com/load_intro.php?id=43, accessed on 1 January 2025), and PharmMapper (https://www.lilab-ecust.cn/pharmmapper, accessed on 1 January 2025). A multifaceted bioinformatics methodology was utilized to investigate prospective therapeutic targets in the context of DN. The initial phase of data extraction involved a comprehensive analysis of several reputable biomedical databases, including GeneCards (https://www.genecards.org, accessed on 1 January 2025), OMIM (https://www.omim.org/, accessed on 1 January 2025), TTD (https://db.idrblab.net/ttd, accessed on 1 January 2025) and DisGeNET (https://disgenet.com, accessed on 1 January 2025), to aggregate genetic information pertinent to disease pathologies. Upon initial identification, candidate genes were subjected to stringent validation and standardization processes involving cross-referencing with the UniProt database to ensure accuracy and consistency. An intersectional analysis between SIN-related and DN-associated targets was conducted using the Venn diagram feature of Venny 2.1.0. To elucidate molecular interactions, a protein–protein interaction network was constructed by integrating data from STRING (https://string-db.org, accessed on 1 January 2025) database queries and employing Cytoscape 3.9.1 for visualization, thereby facilitating the identification of key molecular targets. Biological pathway and functional annotation analyses were conducted utilizing the Bioinformatics platform (https://www.bioinformatics.com.cn/, accessed on 1 January 2025), integrating Gene Ontology classification and KEGG pathway mapping. Molecular docking simulations were performed to evaluate binding affinities and interaction profiles, employing AutoDock 1.5.7 for computational assessment and PyMOL 2.5.0 for structural visualization.

### 4.8. Immunofluorescence

Renal tissue sections were deparaffinized and rehydrated, followed by blocking with 3% bovine serum albumin (BSA) for a duration of 60 min. Subsequently, the sections were incubated with anti-NF-κB antibody (1:500) and anti-IL-1β antibody (1:500) at a temperature of 4 °C for a period of 12 h. After extensive rinsing, the tissue specimens were treated with fluorophore-conjugated secondary antibodies specific to the host species, with incubation maintained at room temperature for 120 min. Nuclear visualization was achieved through brief exposure to a 4′,6-diamidino-2-phenylindole (DAPI) solution. Post-staining, a specialized anti-fade medium was applied to ensure the stability of fluorescence during microscopic examination. Microscopic evaluation was conducted using an Olympus inverted fluorescence imaging system. Image acquisition and processing were performed utilizing the latest version of ImageJ analytical software (version 2.1.0). Quantitative assessment of fluorescence signals was achieved through systematic analysis of multiple microscopic fields.

### 4.9. Immunohistochemistry

Tissue sections were subjected to antigen retrieval by immersion in a citrate-based buffer, followed by incubation in a microwave oven for a duration of 20 min. Upon reaching ambient temperature, the sections were rinsed with phosphate-buffered saline (PBS) and subsequently subjected to hydrogen peroxide treatment to eliminate endogenous peroxidase activity. To mitigate nonspecific interactions, tissue sections were initially incubated in a 3% bovine serum albumin (BSA) blocking solution for a duration of 60 min at ambient temperature. Following the initial preparation, the samples were subjected to incubation with a panel of specific primary antibodies, each directed against pivotal signaling molecules. These included cGAS (1:500), STING (1:200), TBK1 (1:200), and IRF3 (1:200). The incubation of antibodies was conducted at a temperature of 4 °C for a duration of approximately 16 h to facilitate optimal interaction between antigens and antibodies. Post-rinsing, tissue specimens were subjected to incubation with species-specific secondary antibodies for a duration of 60 min at room temperature. Subsequently, chromogenic visualization was achieved through the application of 3,3′-diaminobenzidine (DAB) as the substrate. The stained sections were subsequently subjected to a standardized post-processing protocol, which encompassed nuclear counterstaining using hematoxylin, sequential dehydration through graded ethanol solutions, and permanent mounting in a resinous medium. Microscopic examination revealed the presence of staining, and the mean optical density was quantified utilizing ImageJ software to evaluate the levels of protein expression.

### 4.10. Western Blotting

Total protein was extracted from kidney cortex tissue using RIPA lysis buffer. The collected proteins were separated by sodium dodecyl sulfate-polyacrylamide gel electrophoresis (SDS-PAGE) and transferred onto a polyvinylidene difluoride (PVDF) membrane. The membrane was then incubated at room temperature for 2 h. Primary antibody incubation was carried out at 4 °C for approximately 16 h using the following specific antibodies: rabbit polyclonal anti-cGAS (1:2000), rabbit monoclonal anti-STING (1:1500), mouse monoclonal anti-TBK1 (1:1000), and mouse monoclonal anti-IRF3 (1:1000). After thorough washing, the membranes were incubated with horseradish peroxidase (HRP)-conjugated secondary antibodies (1:5000) at room temperature for 1 h. Following additional washes, protein detection was performed using an enhanced chemiluminescence (ECL) reagent. Protein bands were visualized with the Odyssey Fc Infrared Imaging System, and their intensity was quantified using ImageJ software. Expression levels were normalized to β-actin.

### 4.11. Real-Time Fluorescence Quantitative PCR (RT-qPCR)

Total RNA was extracted from kidney tissues utilizing the RNA extraction kit, followed by an assessment of its purity and concentration. cDNA synthesis was performed using the Takara Bio reverse transcription system. Subsequently, RT-qPCR amplification was executed with the fluorescence detection kit. Gene expression levels were normalized to Gapdh, and relative expression was determined via the comparative Ct (2-ΔΔCt) method. Primer sequences are provided in Table 2.

### 4.12. Statistical Analysis

Data are expressed as mean ± standard deviation (SD). Statistical analyses and figure generation were conducted using GraphPad Prism software (version 10.1.1, San Diego, CA, USA). For normally distributed data, analysis of variance (ANOVA) was performed. If variance homogeneity was satisfied, a *t*-test was applied; otherwise, the Dunnett test was used. When data did not conform to a normal distribution, non-parametric tests were conducted. A *p*-value < 0.05 was considered statistically significant.

## 5. Conclusions

This study, utilizing network pharmacology, molecular docking, and animal experiments, thoroughly explored the mechanism by which SIN inhibits the inflammatory response of DN through the cGAS/STING signaling pathway. We found that SIN significantly inhibits the activation of the cGAS/STING pathway, reduces the expression of NF-κB and downstream inflammatory factors, alleviates renal inflammation, and effectively mitigates the pathological damage of DN. As a natural compound with multi-target effects, SIN potential in the treatment of DN provides new therapeutic insights and directions for the clinical management of this disease.

## Figures and Tables

**Figure 1 pharmaceuticals-18-00934-f001:**
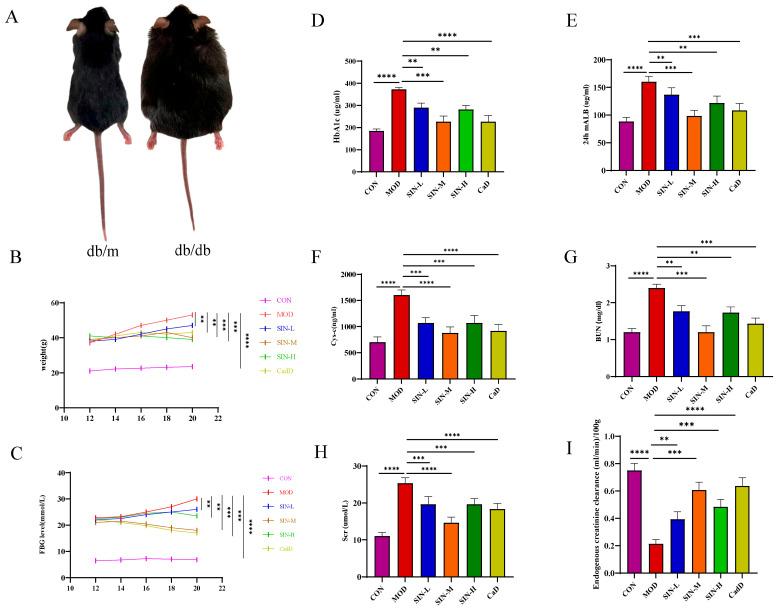
SIN effectively prevents the progression of renal dysfunction in db/db mice. (**A**) Images of db/m and db/db mice. (**B**) Changes in mouse body weight during treatment. (**C**) Changes in fasting blood glucose levels during treatment. (**D**) HbA1c levels in serum. (**E**–I) Levels of 24 h mALB, Cys-c, BUN, Scr and CCr. ** *p* < 0.01; *** *p* < 0.001; **** *p* < 0.0001.

**Figure 2 pharmaceuticals-18-00934-f002:**
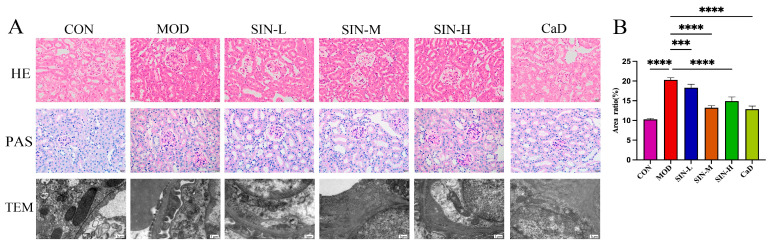
SIN improves renal pathological damage in db/db mice. (**A**) HE staining, PAS staining (magnification: ×400).and TEM images of mouse kidney tissue pathological structure. (**B**) PAS-positive expression analysis. *** *p* < 0.001; **** *p* < 0.0001.

**Figure 3 pharmaceuticals-18-00934-f003:**
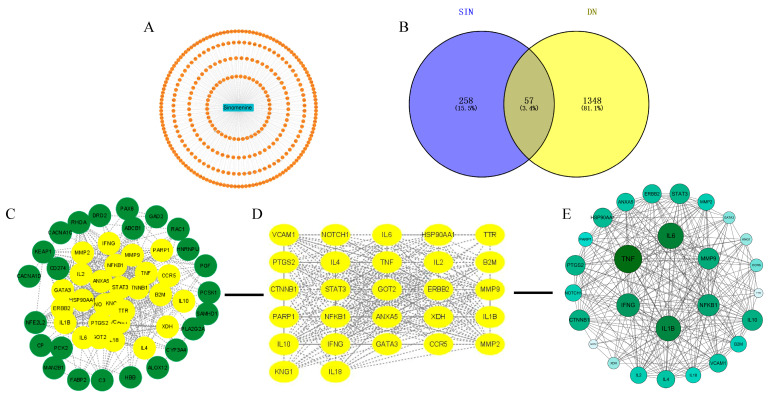
Analysis of SIN and DN-related targets. (**A**) Potential targets of SIN. (**B**) Venn diagram of SIN and DN-related targets. It includes 258 SIN-related targets (left), 1348 DN-related targets (right), and 57 SIN-DN overlapping targets (center). (**C**) Module analysis result of the PPI network. Green nodes and yellow nodes represent different functional modules identified by clustering algorithms, highlighting protein subgroups with close internal interactions. (**D**) Hub gene network within the PPI framework. Nodes represent key hub genes screened based on topological properties, which may play crucial roles in relevant biological processes, and edges show their interaction connections. (**E**) The size of the green circular nodes indicates the degree of association: larger circles represent higher association, and smaller circles represent lower association. The core target is the target in the inner circle.

**Figure 4 pharmaceuticals-18-00934-f004:**
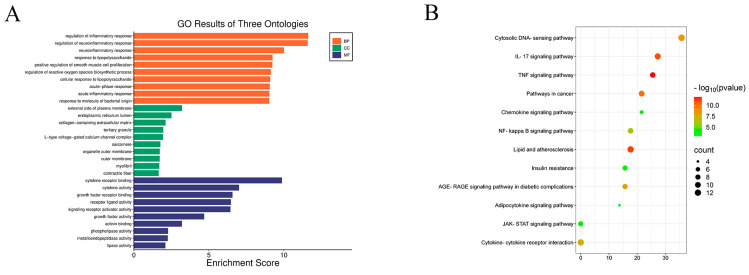
GO and KEGG Enrichment Analysis. (**A**) GO enrichment analysis, with fold enrichment (*y*-axis) and terms (*x*-axis). Orange, green, and purple represent the top 10 core results for BP (biological process), CC (cellular component), and MF (molecular function), respectively. (**B**) KEGG pathway enrichment analysis (DAVID). Pathways (*y*-axis), ratio (*x*-axis), and *p*-value (represented by color change). The size of the bubbles indicates the number of enriched genes in each pathway.

**Figure 5 pharmaceuticals-18-00934-f005:**
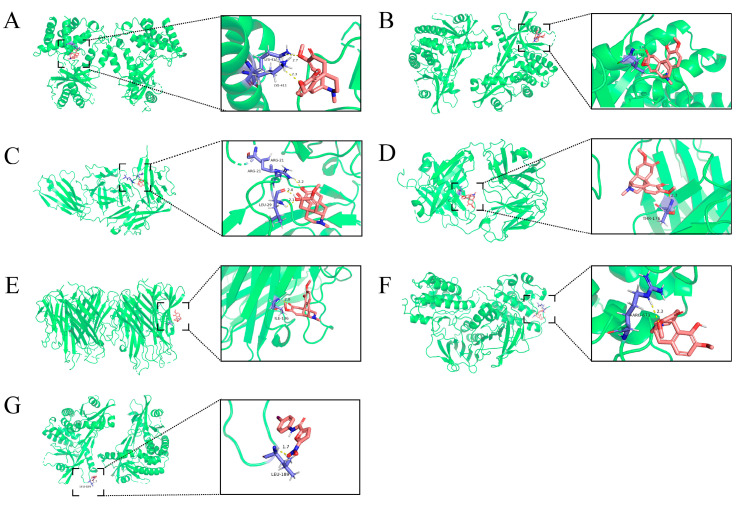
Molecular docking models of SIN with potential core DN targets, Pink represents SIN, and purple represents the target protein. (**A**): SIN-cGAS (−5.1 kcal/mol); (**B**) SIN-STING (−6.2 kcal/mol); (**C**) SIN-IL-1β (−5.9 kcal/mol); (**D**) SIN-IL-6 (−5.4 kcal/mol); (**E**) SIN-TNF-α (−4.5 kcal/mol); (**F**) SIN-NF-κB (−5.4 kcal/mol); (**G**) C-176-STING (−6.5 kcal/mol).

**Figure 6 pharmaceuticals-18-00934-f006:**
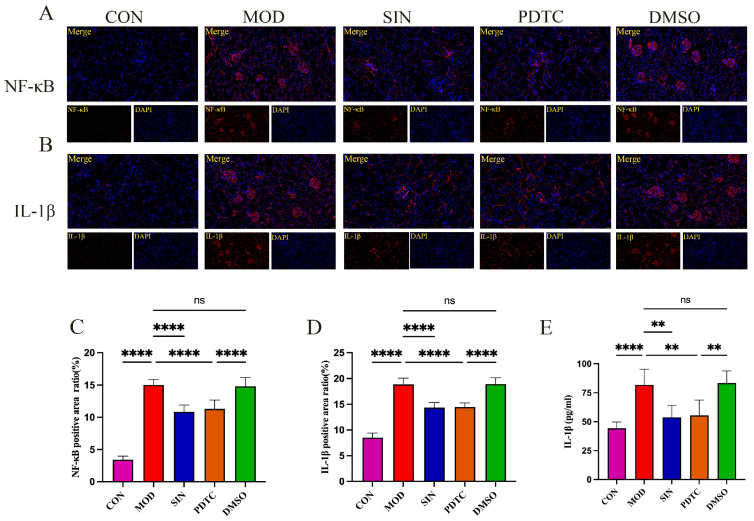
SIN inhibits inflammatory damage in db/db mice. (**A**,**B**) Immunofluorescence of NF-κB and IL-1β in mouse kidney tissue (magnification: ×400). (**C**,**D**) Quantification of fluorescence intensity for NF-κB and IL-1β. (**E**–**H**) Serum levels of IL-1β, IL-6, TNF-α, and MCP-1. (**I**–**K**) Levels of IL-1β, IL-6, and TNF-α in kidney tissue. ** *p* < 0.01; *** *p* < 0.001; **** *p* < 0.0001; ns, not significant.

**Figure 7 pharmaceuticals-18-00934-f007:**
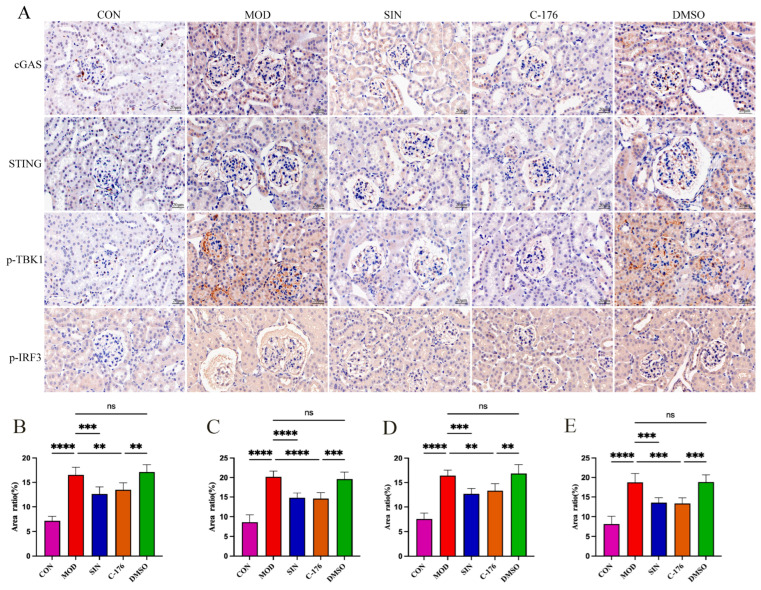
(**A**) Immunohistochemistry of cGAS, STING, TBK1, and IRF3 (magnification: ×630); (**B**–**E**) ImageJ (version 2.1.0) analysis of the positive area ratio for cGAS, STING, TBK1, and IRF3 immunohistochemistry. ** *p* < 0.01; *** *p* < 0.001; **** *p* < 0.0001; ns, not significant.

**Figure 8 pharmaceuticals-18-00934-f008:**
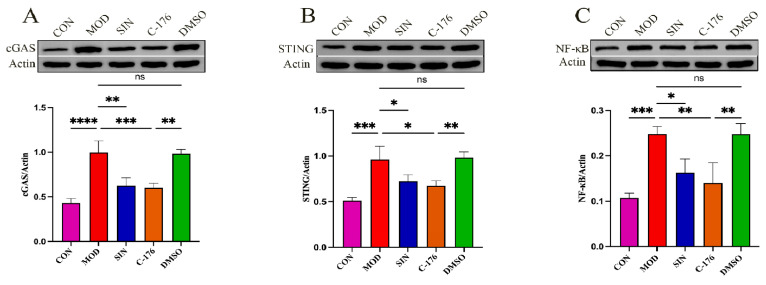
(**A**–**E**) Western blotting analysis showing protein levels of cGAS, STING, NF-κB, TBK1, and IRF3 in kidney tissue, along with corresponding grayscale values; (**F**–**I**) RT-qPCR analysis of mRNA expression of cGAS, STING, TBK1, and IRF3 in db/db mice; * *p* < 0.05, ** *p* < 0.01; *** *p* < 0.001; **** *p* < 0.0001; ns, not significant.

**Table 1 pharmaceuticals-18-00934-t001:** The CKD-PIS was used to score each group.

Scores Groups	CON	MOD	SIN-L	SIN-M	SIN-H	CaD
glomerular sclerosis	0	2	1	1	2	1
Tubular atrophy	0	2	2	1	2	0
Inflammatory cell infiltration	0	2	1	0	0	2
Overall score	0	6	4	2	4	3

**Table 2 pharmaceuticals-18-00934-t002:** Primer sequences used for qRT-PCR analysis (Mouse).

Primer Name	Forward Primer (5′-3′)	Reverse Primer (3′-5′)
M-cGAS	CACGAGGAAATCCGCTGAGTC	CACGAGGAAATCCGCTGAGTC
M-TBK1	GCAGTGCTAAGAAAGGACCATCA	TGCCTGAAGACCCTGAGAAAGAC
M-IRF3	CTACGGCAGGACGCACAGAT	GCAGCTAACCGCAACACTTCT
M-STING	TCGGGTTTATTCCAACAGCG	GTTTAGCCTGCTCAAGCCGAT
M-GAPDH	CCTCGTCCCGTAGACAAAATG	TGAGGTCAATGAAGGGGTCGT

## Data Availability

The data presented in this study are available on request from the corresponding author. The data are not publicly available due to privacy.

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
