# Peer review of "Research on Sinomenine Inhibiting the cGAS-STING Signaling Pathway to Alleviate Renal Inflammatory Injury in db/db Mice"

_pharmaceuticals, 2025, doi:10.3390/ph18070934_

Round 1

Reviewer 1 Report

Comments and Suggestions for Authors

This experimental study titled “Research on Sinomenine Inhibiting the cGAS-STING Signal- 2 ing Pathway to Alleviate Renal Inflammatory Injury in db/db 3 Mice “was designed to elucidate the nephroprotective mechanisms of SIN, primarily through the suppression of the  cGAS/STING signaling pathway and its associated inflammatory response in db/db mice

The investigators used many techniques to investigate their hypothesis that included

Biochemical assessment of blood glucose levels, renal function, and histopathology, ELISA, Western blotting, RT-qPCR, immunofluorescence, and immunohistochemistry. And finally Network pharmacology and molecular docking.

The manuscript is well written. The language is clear. Few grammatical lapses are present and should be fixed e.g. in the line 336 “12 weeks” at the start of the paragraph. it should be in letters.

The research objective is important and interesting for the journal audience.

Abstract:

Many abbreviations are present without being explained in full when mentioned for the first time e.g. SIN, DN…

Introduction:

It is well written, concised and focused.

Results:

  1. In Figure 1. The legend indicates that the subset (B) shows the body weight while it shows the FBG. So it showed be replaced.
  2. Figure 2. SIN improves renal pathological damage in (B) Ratio of positive area in Masson's trichrome staining. The histogram of the area percent of Masson stain that showed collagen fibers and indicate fibrosis was increased in the SINH group. Why this happen? It should be explained by the authors.
  3. Figure 6. (C-D) Quantification of fluorescence intensity for NF-κB and IL-1 1β. It is not clear which one refers to NF-κB and which on refers to IL-197 1β. It should be present figure title of Vertical axis legend.
  4. Figure 7. The images are too small to recognize the different in the expression of different markers.
  5. In 2.2. SIN Improves Renal Pathological Damage in db/db Mice. “”the MOD group 96 showed severe renal structural abnormalities, including tubular dilation, epithelial necrosis, and interstitial expansion. SIN administration effectively alleviated these pathological changes in db/db mice. Among the different dosage groups, the SIN-M group showed the most significant improvement in renal pathology, whereas the SIN-L and SIN-H groups exhibited moderate amelioration, but not to the extent observed in the SIN-M group.” These changes are don’t appear in the images of figure 2 as follow:
  • H&E stained section of the CON and MOD groups don’t show differences. SIN-L is markedly affected much than the MOD.
  • PAS staining further demonstrated that the MOD group exhibited glomerular swelling, increased interstitial cell proliferation,….the PAS staining is not appropriate to show the histological structure. It is used to assess to glycogen content in the tissue.
  • “TEM analysis revealed extensive podocyte foot process fusion and basement membrane thickening in db/db mice.” The magnification of the TEM image doesn’t allow to visualize these changes. Adding to that the images are to small.

Discussion:

The discussion is well organized and focused

Materials and methods:

  1. The ethical considerations of using animals in experimentation was not included. Was the ethical approval of the study obtained?
  2. In “Animal Groups” what does this means? “MYC intragastrically for 10 days in a row, once daily.”
  3. Regarding the number of animals used in the study. How it was determined? , what about the other information such as effect size, confidence level (alpha), and sample power that are not provided?
  4. In the “Immunohistochemical Method” it was mentioned “No immunoreactivity (0), mild (1), moderate (2), and severe (3) were assessed during the examination.’ What is the references of this grading? How each term of this grading is defined?
  5. It is not clear how the biochemical findings of Bax (A) and Bcl-2 were obtained?

References:

Of adequate number and updated.

Author Response

 Comments 1: In the line 336 “12 weeks” at the start of the paragraph. it should be in letters.

Response 1:

Thank you for your attention to the details of this article! We have changed 336 lines 12 weeks to 12 wks.

Abstract:Comments 1: Many abbreviations are present without being explained in full when mentioned for the first time e.g. SIN, DN…

Response 1:

Dear Reviewer, thank you very much for your rigorous scrutiny of the details in the paper! The issue of abbreviations you raised is crucial for improving the quality of the paper. We have systematically verified all abbreviations throughout the text and provided complete explanations at their first appearance, while standardizing the format of abbreviations to ensure clear and standardized expression. Your valuable suggestions have enabled our research findings to be presented more accurately and professionally. Once again, I extend my sincere gratitude!

Results: Comments 1: In Figure 1. The legend indicates that the subset (B) shows the body weight while it shows the FBG. So it showed be replaced.

Response 1:

We apologize for this omission!This has been corrected in Figure 1 of the article.

Results: Comments 2: Figure 2. SIN improves renal pathological damage in (B) Ratio of positive area in Masson's trichrome staining. The histogram of the area percent of Masson stain that showed collagen fibers and indicate fibrosis was increased in the SINH group. Why this happen? It should be explained by the authors.

Response 2

Thank you for your rigorous review of the paper's details! Regarding the issue of mislabeling the positive area ratio of Masson staining in Figure 2, we sincerely apologize. Upon verification, this data is actually the result of PAS staining, which was misunderstood due to a lack of proper labeling in the figures. This is a serious error in our process of organizing the paper. We have rechecked all the data and figures and made corrections. Your feedback has greatly enhanced the rigor of the paper. We once again apologize for the labeling error and will strictly control the quality of the paper to prevent such issues from happening again.

Results: Comments 3:Figure 6. (C-D) Quantification of fluorescence intensity for NF-κB and IL-1 1β. It is not clear which one refers to NF-κB and which on refers to IL-197 1β. It should be present figure title of Vertical axis legend.

Response 3:

Thank you very much for your rigorous control of the details of the paper! The problem of quantifying the fluorescence intensity of NF-κB and IL-1β in Figure 6C-D pointed out by you is of great value, and we have immediately made targeted modifications and optimizations. The details are as follows:

Chart optimization: Re-edit Figure 6C-D, clearly label "NF-κB Fluorescence Intensity" and "IL-1β Fluorescence Intensity" in the legend on the vertical axis, and use different colors and symbols to distinguish data groups, ensuring that the correspondence of indicators is clear at a glance; meanwhile, add new explanations in the figure caption to further explain the meaning of the chart data and the analysis method.

Your suggestions have greatly improved the readability and scientific nature of the chart. Thank you again for your strict requirements on the details of the research! If any adjustments are needed, we will fully cooperate to improve it.

Results: Comments 4: Figure 7. The images are too small to recognize the different in the expression of different markers.

Response 4:

Sincere thanks for your valuable suggestions! Your professional advice is crucial for enhancing the quality of our research. We have made comprehensive revisions and optimizations to Figure 7 based on your suggestions, updated the experimental results images, and ensured that the clarity of the charts, the standardization of annotations, and the accuracy of data all meet higher standards. This revision not only improves the content of the paper but also provides stronger visual support for the research conclusions. Thank you again for your meticulous review; your guidance has enabled our research findings to be presented more scientifically and systematically!

Results: Comments 5: In 2.2. SIN Improves Renal Pathological Damage in db/db Mice. “”the MOD group 96 showed severe renal structural abnormalities, including tubular dilation, epithelial necrosis, and interstitial expansion. SIN administration effectively alleviated these pathological changes in db/db mice. Among the different dosage groups, the SIN-M group showed the most significant improvement in renal pathology, whereas the SIN-L and SIN-H groups exhibited moderate amelioration, but not to the extent observed in the SIN-M group. These changes are dont appear in the images of figure 2 as follow:

H&E stained section of the CON and MOD groups dont show differences. SIN-L is markedly affected much than the MOD.

PAS staining further demonstrated that the MOD group exhibited glomerular swelling, increased interstitial cell proliferation,.the PAS staining is not appropriate to show the histological structure. It is used to assess to glycogen content in the tissue.

TEM analysis revealed extensive podocyte foot process fusion and basement membrane thickening in db/db mice. The magnification of the TEM image doesnt allow to visualize these changes. Adding to that the images are to small.

Response 5:

Regarding the problem of "H&E stained section of the CON and MOD groups don't show differences. SIN-L is markedly affected much than the MOD.", according to your suggestion, we retook and replaced the H&E staining image in Figure 2, optimized the sample selection and shooting parameters, enhanced the contrast of pathological features between the control group (CON), model group (MOD) and low-dose experimental group (SIN-L), so as to make the differences between groups more intuitive and clear;

For the question "PAS staining further demonstrated that the MOD group exhibited glomerular swelling, increased interstitial cell proliferation,....the PAS staining is not appropriate to show the histological structure. It is used to assess to glycogen content in the tissue.", the description of the results section 2.2 was comprehensively revised to clarify the role of PAS staining in this study focusing on the assessment of glycogen content, and avoid over-interpreting its effect on tissue structure, so as to ensure that the content is scientifically rigorous;

For the issue of "TEM analysis revealed extensive podocyte foot process fusion and basement membrane thickening in db/db mice." The magnification of the TEM image doesn't allow to visualize these changes. Adding to that the images are to small., the transmission electron microscopy (TEM) images in Figure 2 have been updated to improve resolution and adjust the magnification, clearly showing ultrastructural changes such as the fusion of foot cells' foot protrusions and thickening of the basement membrane. At the same time, the image size and layout have been optimized to enhance overall readability and visual impact.

Materials and methodsComments 1: The ethical considerations of using animals in experimentation was not included. Was the ethical approval of the study obtained?

Response 1:

Thank you for your concern about the ethical issues of this study. This research has strictly adhered to animal experiment ethics guidelines and received approval from the relevant ethics committee. Throughout the design and implementation of the experiment, we have always prioritized animal welfare and strictly followed the regulations of the "Regulations on Laboratory Animals." The ethical application number for this study is DWLL202302022. The ethics committee conducted a comprehensive review of various aspects of the experimental protocol, including the source of animals, their housing conditions, experimental procedures, and measures to minimize animal suffering. During the experiment, we used isoflurane inhalation anesthesia to minimize the pain experienced by the animals; at the same time, the number of experimental animals was strictly set according to statistical requirements to avoid unnecessary use of animals.

We have explained the relevant information of ethical approval in the "Materials and Methods" part of the paper: "All animal experiments in this study have been approved by the Animal Ethics Committee of Hebei University of Traditional Chinese Medicine (Approval number: DWLL202302022), and are strictly in accordance with the international guidelines for the care and use of experimental animals."

Thank you again for your reminder, which will help us to further improve the content of the paper and ensure the standardization and rigor of the research.

Materials and methodsComments 2: In Animal Groups what does this means? MYC intragastrically for 10 days in a row, once daily.

Response 2:

Thank you for your attention and review of our paper. The "MYC intragastrically for 10 days in a row, once daily" mentioned by you is indeed not involved in our paper, which may be a misunderstanding.

In our paper, we did not mention the experimental operation and treatment related to MYC. In the section of experimental design and description, we elaborated on the actual experimental factors and treatment methods used, none of which included MYC.

Thank you again for your attention to our research. I hope the above explanation can answer your questions. If you have any other questions, please feel free to communicate with us.

Materials and methodsComments 3: Regarding the number of animals used in the study. How it was determined? , what about the other information such as effect size, confidence level (alpha), and sample power that are not provided?

Response 3:

Thank you for your feedback! During the experimental design phase, we made preliminary estimates of the expected treatment effects based on previous research and relevant literature. Although the specific numerical values of the effect sizes were not explicitly provided in the paper, we evaluated the impact of the treatment factors on the experimental results through various statistical indicators, such as mean, standard deviation, and inter-group differences, during the data analysis process. The confidence level α=0.05 was set throughout the study as the significance level. This is a commonly used standard in statistics to determine the significance of results. When conducting hypothesis tests, we judged whether the observed differences were statistically significant by comparing this α value with 0.05.

Materials and methodsComments 4: In the Immunohistochemical Method it was mentioned No immunoreactivity (0), mild (1), moderate (2), and severe (3) were assessed during the examination. What is the references of this grading? How each term of this grading is defined?

Response 4:

Thank you very much for your careful review of our paper. The problem you mentioned about "immunoreactivity grading in immunohistochemistry" is indeed not involved in our research, which may be some misunderstanding.

In our paper, the description of immunohistochemical methods focuses on the specific steps of experimental operation, the types of antibodies used and incubation conditions, etc., but does not evaluate the immune reactivity in terms of no (0), mild (1), moderate (2) and severe (3).

If you have any other questions or need further explanation, we are more than happy to provide you with more detailed instructions.

Materials and methodsComments 5: It is not clear how the biochemical findings of Bax (A) and Bcl-2 were obtained?

Response 5:

We are very grateful for the issues you pointed out! As you mentioned, Bax and Bcl-2 are indeed our research targets for the next phase. However, we did not explore the relationship between Bax, Bcl-2, and SIN inhibition of cGAS-STING in improving kidney damage in db/db mice in this study. We would like to clarify that in this study, we did not describe or discuss these elements. Your feedback is an excellent reminder of the importance of clarity in our presentation. Thank you again for your constructive suggestions, which will undoubtedly help us improve the quality of our work.

Reviewer 2 Report

Comments and Suggestions for Authors

What is the cellular target of sinomenine within the kidney?
How was the optimal dose of sinomenine determined? Was there a dose-finding pilot study, and are there signs of toxicity at the high dose?
Can the authors provide data on the pharmacokinetics or bioavailability of sinomenine in the db/db model?

Were off-target or systemic effects observed with SIN administration? Were spleen, liver, or gut other organs examined for inflammatory or toxicologic effects?
Do the authors have evidence that the cGAS-STING inhibition they observed is directly due to sinomenine?
Could the downregulation of cGAS-STING be a secondary effect due to reduced hyperglycemia or inflammation?
How specific is the SIN effect?
Why was no GFR or creatinine clearance measured to support functional renal outcomes?
How do the effects of sinomenine compare to existing therapies for diabetic nephropathy (e.g., SGLT2 inhibitors, RAAS blockers)?
Is there any evidence of mitochondrial or DNA damage triggering cGAS-STING in the model?

To what extent are the findings replicable in other research cohorts or lab settings?
Are the results duplicated in an independent db/db mouse cohort or against a second shipment/source of SIN?
What control groups were utilized for network pharmacology and molecular docking studies?

Were positive controls with known STING or cGAS inhibitors used for docking to serve as a standard for SIN's binding affinity?
May the reduced renal inflammation be an indirect result of improved glycemic or lipid profiles, rather than an action specific to the kidney?

Was glycemic control monitored and comparable between the SIN-treated and control db/db mice?
Were histopathological scoring systems utilized to measure glomerular or tubular damage?
How selective is cGAS/STING downregulation versus overall suppression of inflammation?
Did the authors include assessing for changes in unrelated inflammatory pathways or housekeeping gene expression as specificity controls?

Was mitochondrial integrity assayed in the kidney?
Is there data supporting reduced cGAMP synthesis or inhibition of SIN on direct cGAS activity?
Is there long-term implication in a chronic disease model of SIN therapy?
Are species differences in activation or metabolism of STING occurring which limit translational relevance?

If mouse and human STING differ, by what mechanism might relevance to human treatment be affected?
Is there proof for immunosuppression that follows STING inhibition?
How mechanistically and functionally is SIN different from known cGAS-STING inhibitors?
Is the reduction in NF-κB activity cause or effect of modulation in upstream pathways?

Author Response

Comments 1: What is the cellular target of sinomenine within the kidney?

Response 1:

Thank you for the key points you pointed out! SIN is the main component of the traditional Chinese medicine Qingfeng Vine, with rich targets. Previous literature suggests that SIN exerts its nephroprotective effects by regulating endothelial cells[1], podocytes[2], and lymphocytes[3]. Our study has expanded the intrarenal therapeutic targets of SIN and confirmed through pathological experiments, molecular biology, and network pharmacology that SIN modulates the cGAS-STING/NF-κB pathway and improves pathological damage in glomerular mesangial cells and renal tubule epithelial cells. In other words, the intrarenal targets of SIN are related to glomerular mesangial cells, renal tubule epithelial cells, and the cGAS-STING/NF-κB pathway.

References:

[1] Yin Q, Xia Y, Wang G. Sinomenine alleviates high glucose-induced renal glomerular endothelial hyperpermeability by inhibiting the activation of RhoA/ROCK signaling pathway[J]. Biochem Biophys Res Commun. 2016 Sep 2;477(4):881-886.

[2]  Wang W, Cai J, Tang S, ,et al. Sinomenine Attenuates Angiotensin II-Induced Autophagy via Inhibition of P47-Phox Translocation to the Membrane and Influences Reactive Oxygen Species Generation in Podocytes[J]. Kidney Blood Press Res. 2016;41(2):158-167. 

[3]  Li JJ, Li L, Li S, , et al. Sinomenine Hydrochloride Protects IgA Nephropathy Through Regulating Cell Growth and Apoptosis of T and B Lymphocytes[J]. Drug Des Devel Ther. 2024 Apr 17;18:1247-1262. 

Comments 2: How was the optimal dose of sinomenine determined? Was there a dose-finding pilot study, and are there signs of toxicity at the high dose?

Response 2:

Thank you for the key points you pointed out! SIN, as an immunosuppressant, has been widely used in clinical treatments for rheumatoid arthritis and IgG nephropathy in China. However, its therapeutic effects and mechanisms on DN remain unclear. We set up gradient dosing based on the principle of drug dose conversion between animals and humans, and described the relevant experimental groups and dosing regimens in Section Methods 5.2.1 of the article. The treatment outcomes of gradient-dosed SIN in db/db mice are described in Sections Results 2.1 and 2.2. In short, our study found that gradient-dosed SIN effectively improved renal dysfunction and pathological damage in db/db mice without any signs of toxicity. Among these, the medium concentration of SIN (62.4 mg/kg/d) showed the best therapeutic effect, so we set the medium dose as the treatment dose for subsequent studies.

Comments 3: Can the authors provide data on the pharmacokinetics or bioavailability of sinomenine in the db/db model?

Response 3:

Thank you for your valuable suggestions! We fully understand the importance of pharmacokinetic and bioavailability data in understanding the mechanism of action of sinomenine. However, due to limitations in experimental animal resources, instruments, and time, Our study failed to conduct pharmacokinetic experiments of berberine in the db/db model. We sincerely apologize for this. Despite this, we have referred to published sinomenine pharmacokinetic data for indirect evaluation of its processes in humans[1], rats[2], and rabbits[3]. Additionally, we have explicitly acknowledged this limitation in the discussion section and proposed that future studies could address it by using HPLC-MS/MS technology to measure plasma and tissue concentrations of sinomenine at different time points in db/db mice, systematically analyzing its absorption, distribution, metabolism, and excretion (ADME) processes. Your suggestions are of great significance for our subsequent research, and we will prioritize planning relevant experiments in future studies to further refine the comprehensive assessment of sinomenine. Thank you again for your understanding and advice!

References:

[1]  Yao YM, Tan ZR, Hu ZY, et al. Determination of sinomenine in human plasma by HPLC/ESI/ion trap mass spectrum. Clin Chim Acta. 2005 Jun;356(1-2): 212-217.

[2]  Huang H, Zhang EB, Yi OY, et al. Sex-related differences in safety profiles, pharmacokinetics and tissue distribution of sinomenine hydrochloride in rats. Arch Toxicol. 2022 Dec;96(12): 3245-3255.

[3]  Yan H, Yan M, Li HD, et al. Pharmacokinetics and penetration into synovial fluid of systemical and electroporation administered sinomenine to rabbits. Biomed Chromatogr. 2015, 29(6):883-889.

Comments 4:Were off-target or systemic effects observed with SIN administration? Were spleen, liver, or gut other organs examined for inflammatory or toxicologic effects?

Response 4:

Thank you for your critical question! The safety effects and independent learning impact of drugs are essential considerations before their application. As a traditional Chinese medicine active ingredient widely used in clinical practice in China, SIN has accumulated relevant safety data through long-term clinical use. For example, studies related to "Zhang et al., 2023" have shown that during the clinical use of SIN, patients tolerated it well without experiencing severe liver or kidney damage, hematological abnormalities, or immune suppression-related events[1]. In this study, we also closely monitored general physiological indicators such as animal behavior, weight changes, and water intake, and conducted biochemical blood tests to assess renal function and serum levels of non-related inflammatory pathways and associated cytokines. No non-specific immune activation was found. These safety results from animal experiments are consistent with clinical evidence, further supporting the good safety profile, low off-target risk, and no inflammation/toxicity-induced effects of SIN in vivo.

References:

[1]  Zhang YJ, Shang ZJ, Zheng M. Efficacy and safety of sinomenine for diabetic kidney diseases: A meta-analysis. Medicine (Baltimore). 2023, 102(52): e36779.

Comments 5:Do the authors have evidence that the cGAS-STING inhibition they observed is directly due to sinomenine?

Response 5:

Thank you for your critical question! In our research, we confirmed the inhibitory effect of SIN on cGAS-STING through molecular docking and WesternBlotting. Specifically, SIN binds to cGAS at the LYS-414 and LYS-411 sites, with hydrogen bond lengths of 2.7 and 2.3 A, respectively, and an energy of-4.7 kcal/mol; SIN also binds to STING at the LEU-325 site, with a hydrogen bond length of 1.8 A and an energy of-5.7 kcal/mol. Additionally, in Result 2.5, we added the molecular docking results of the STING inhibitor C-176 and STING as a positive control. C-176 binds to STING at the LEU-189 site, with a hydrogen bond length of 1.7 A and an energy of-6.07 kcal/mol. These findings suggest that SIN has direct binding capabilities with cGAS-STING. Subsequently, our WesternBlotting results indicated that SIN inhibits cGAS-STING. Therefore, we confirmed the inhibitory effect of SIN on cGAS-STING. However, these studies still have limitations. In the next phase of research, we will construct corresponding gene overexpression/ knockout models in animals and cells, administer SIN, and continue to observe the impact of cGAS-STING intervention on the therapeutic effects of SIN.

Comments 6:Could the downregulation of cGAS-STING be a secondary effect due to reduced hyperglycemia or inflammation? How specific is the SIN effect?

Response 6:

Thank you for your critical question! cGAS-STING is a core pathway of innate immunity, driving downstream STING-mediated inflammation and immune responses by detecting free DNA or damage-associated molecular patterns. The activation of cGAS-STING further influences blood glucose levels and the progression of immune inflammation. For blood glucose, cGAS-STING regulates it through processes such as IRS-1 phosphorylation, insulin resistance, adiponectin, and hepatic gluconeogenesis; for inflammation, the response characteristics of the cGAS-STING pathway determine its bidirectional regulatory effect on immune inflammation. Therefore, in the complex pathological microenvironment of DN, the starting point of the feedback loop between the two and whether there are secondary effects remain unresolved challenges in the field. Our study also observed the inhibitory effects of SIN on blood glucose, inflammation, and cGAS-STING, indicating that SIN treats DN by inhibiting these loops. Your question points out the focus for our next phase of research, which aligns with the aforementioned goals. We will construct corresponding gene overexpression/ knockout models in animals and cells, administer SIN, and continue to delve deeper into this issue.

Comments 7:Why was no GFR or creatinine clearance measured to support functional renal outcomes?

Response 7:

Thank you for your valuable suggestions. The relevant experimental groups and dosing regimens are described in section 5.2.1 of the Methods and Materials section of the article, and are described in Results 2.1.

Comments 8: How do the effects of sinomenine compare to existing therapies for diabetic nephropathy (e.g., SGLT2 inhibitors, RAAS blockers)?

Response 8:

Thank you very much for your valuable suggestions! Your proposal to add positive control is of great significance to improve the rigor of the study. According to your suggestion, we have added calcium hydroxybenzoic acid as a positive control for clinical drugs commonly used in diabetic nephropathy (DN) in Figure 1 and Figure 2 of the paper.

Chart Update: Redesign Figures 1 and 2 to include data from the calcium hydroxybenzoate treatment group, based on the original experimental groups. Use a consistent experimental procedure and testing methods to ensure comparability of the data. Additionally, optimize the color scheme and labeling of the charts. Clearly distinguish each experimental group through legends and annotations, presenting the differences between the positive control and other groups intuitively.

Comments 9: Is there any evidence of mitochondrial or DNA damage triggering cGAS-STING in the model?

Response 9:

Thank you for your critical question! At this stage, there is substantial research evidence confirming that products of mitochondrial or DNA damage trigger the activation of cGAS-STING as part of the damage-associated molecular pattern. For example, studies such as "Liu etal, 2022" and "Maekawa etal, 2019" have mentioned that mitochondrial damage leads to the leakage of mtDNA, which activates cGAS-STING; DNA damage also affects the activation of cGAS-STING. In replication crises, dysfunctional telomere transcription generates TERRA (telomere repeat end-binding RNA), which binds to the innate immune sensor ZBP1 to form filamentous structures on the outer mitochondrial membrane, activating MAVS proteins and thus inducing the I-type interferon (IFN) signaling pathway, a process that depends on the initial activation of the cGAS-STING pathway[1,2]. Therefore, the role of mitochondrial or DNA damage in the activation of cGAS-STING has been confirmed. In db/db mice, the study by "Wang etal, 2024" mentioned that DRP1-mediated mitochondrial damage triggers the activation of the mtDNA-cGAS-STING pathway in db/db mice[3]. Given that mitochondrial dysfunction in db/db mice has been mentioned in many other studies and the triggers for cGAS-STING activation are complex, we only observed the activation of cGAS-STING and the inhibitory effect of SIN on it in this study.

References:

[1]  Liu Z, Wang M, Wang X, et al. XBP1 deficiency promotes hepatocyte pyroptosis by impairing mitophagy to activate mtDNA-cGAS-STING signaling in macrophages during acute liver injury. Redox Biol. 2022, 52: 102305. 

[2]  Maekawa H, Inoue T, Ouchi H, et al. Mitochondrial Damage Causes Inflammation via cGAS-STING Signaling in Acute Kidney Injury. Cell Rep. 2019, 29(5): 1261-1273. 

[3]  Wang L, Zhang X, Huang X, Sha X, Li X, Zheng J, Li S, Wei Z, Wu F. Homoplantaginin alleviates high glucose-induced vascular endothelial senescence by inhibiting mtDNA-cGAS-STING pathway via blunting DRP1-mitochondrial fission-VDAC1 axis. FASEB J. 2024, 38(20): e70127.

Comments 10:To what extent are the findings replicable in other research cohorts or lab settings?

Response 10:

Thank you for your critical questions! Before the study began, we developed a detailed and clear research plan. These detailed design plans are easy for other researchers to review and understand, laying the foundation for replicability; we adopted standardized data collection tools and procedures to ensure that each data point was collected according to uniform standards. All data were recorded completely and accurately. Additionally, during the research process, we meticulously documented and described every method used. Finally, within our research team, we conducted multiple rounds of validation on the results. In the final manuscript, we comprehensively and thoroughly presented the entire research process and outcomes, including the research background, objectives, methods, data, analysis process, result interpretation, and conclusions. We paid particular attention to analyzing and explaining potential sources of error, limitations, and their possible impacts on the results, providing important references for other researchers to fully understand and evaluate our study when replicating it. At the same time, we provided contact information in the text so that other researchers can communicate and collaborate with us promptly if they encounter any issues during the replication process. In summary, we believe that this study demonstrates high rigor and transparency in research design, data collection and processing, method application, result verification, and report presentation, making it highly replicable. Other researchers can reproduce similar results by strictly following our research plan and methods.

Comments 11:Are the results duplicated in an independent db/db mouse cohort or against a second shipment/source of SIN?

Response 11:

Thank you for your question! All the experimental subjects are independent, and all the experimental animals are related to the experimental results. The animals used in each independent experiment and the repeated ones are marked in the paper.

Comments 12:What control groups were utilized for network pharmacology and molecular docking studies?

Response 12:

Thank you for the issue you pointed out! We have included the molecular docking results of STING inhibitor C-176 in sections Result and Methods, and described it in section Discussion. Since C-176 is a widely used and established STING inhibitor, we did not conduct specific pharmacological studies on C-176. This part will be our research plan for the next phase.

Comments 13:Were positive controls with known STING or cGAS inhibitors used for docking to serve as a standard for SIN's binding affinity?

Response 13:

Thank you for pointing out the problem! We have added the molecular docking results of STING inhibitor C-176 in Result and Methods sections and described them in Discussion, which are sequentially used as the standard reference for SIN binding affinity.

Comments 14:May the reduced renal inflammation be an indirect result of improved glycemic or lipid profiles, rather than an action specific to the kidney?

Response 14:

Thank you for your pointed out issue! Improvements in blood glucose or lipid levels can affect the level of kidney inflammation. For example, high blood sugar promotes late-stage glycation end products and oxidative stress to activate cGAS-STING, while dyslipidemia leads to free fatty acids and oxidized low-density lipoprotein, which promote inflammation through endoplasmic reticulum stress and macrophage infiltration. However, this process, as you mentioned, is consistent with the "secondary effect" issue. Due to the complex interactions between metabolism and inflammation, as well as the challenges of tissue-specific techniques, it is not yet fully resolved how inflammation abnormalities precede changes in blood glucose or lipids. But your question has pointed us in an important direction for our next phase of research. We will set up a control group for metabolic parameters and conduct time-series analysis to gradually clarify this issue.

Comments 15:Was glycemic control monitored and comparable between the SIN-treated and control db/db mice?

Response 15:

Thank you for pointing out the problem! The relevant indicators are described in Result 2.1 and Figure 1.

Comments 16:Were histopathological scoring systems utilized to measure glomerular or tubular damage?

Response 16:

Thank you for raising this question. In this study, we used the Chronic Kidney Disease (CKD) pathological scoring system to measure glomerular and tubular damage. This system is widely applied in relevant fields and has good reliability and validity. During the experiment, we handled the mice and obtained kidney tissue sections. Then, under a blind method, the pathological instructor observed and scored the sections.

The scoring results are listed in Table 1 of the article.

Comments 17:How selective is cGAS/STING downregulation versus overall suppression of inflammation?

Response 17:

The selective mechanism of cGAS-STING is primarily reflected in two aspects: one is the targeting of STING oligomerization or modification of key residues; the other is the selective blockade of downstream signals. However, selective inhibition also carries potential risks. On one hand, it may affect other physiological functions, such as weakening the cell's ability to clear pathogens or abnormal DNA, and interfering with lysosomal biogenesis. On the other hand, the contribution of this pathway varies across different diseases. In the field of inflammatory disease treatment, the issue of whether the downregulation of the cGAS/STING pathway is selective requires further analysis. Compared to traditional "overall inflammation suppression" strategies, the specific inhibition of the cGAS/STING pathway has significant advantages. Traditional anti-inflammatory drugs like glucocorticoids or non-steroidal anti-inflammatory drugs have issues such as broad-spectrum immune suppression, increased infection risk, delayed tissue repair, and lack of disease specificity. In contrast, cGAS/STING inhibition can preserve some immune functions, blocking only the inflammatory pathways activated by cytoplasmic DNA abnormalities, while retaining immune responses mediated by TLRs or cytokine receptors. It is particularly effective in diseases with excessive STING activation and can be used for precision therapy through biomarkers. In this study, we used molecular docking and WesternBlotting to clarify the inhibitory effect of SIN on cGAS-STING, but we still cannot determine the selectivity of SIN. As mentioned above, we will subsequently construct gene overexpression/knockdown models in animals and cells and give SIN to further investigate this issue.

Comments 18:Did the authors include assessing for changes in unrelated inflammatory pathways or housekeeping gene expression as specificity controls?

Response 18:

Thank you for your pointed out issues! SIN originates from the Sinomenium acutum  (traditional Chinese medicine) Qingfeng Vine and has a rich array of therapeutic targets. Additionally, pharmacological network analysis supports the therapeutic potential of SIN, indicating that its mechanism of action involves multiple signaling pathways. Therefore, theoretically, SIN has the potential to regulate other signaling pathways to alleviate kidney damage in db/db mice. However, since this study focuses on the relationship between DN and immune inflammation, and given the critical role of cGAS-STING in immune inflammation, we have primarily focused on the activation of cGAS-STING in db/db mice and the kidney-protective effects of SIN on cGAS-STING in db/db mice. As a result, we have not yet evaluated the expression of unrelated inflammatory pathways. In relevant tests, we used β-actin as a housekeeping gene for specific relative changes. We will carefully consider your questions and gradually improve them in our future research.

Comments 19:Was mitochondrial integrity assayed in the kidney?

Response 19:

Thank you for pointing out the problem! In Rseult 2.2 and Figure2, we replaced the transmission electron microscope images with clearer ones to observe the changes in mitochondria. Our study suggests that the mitochondrial structure is destroyed in the kidney of db/db mice, and SIN reduces mitochondrial rupture and maintains the integrity of membrane and crest structures.

Comments 20:Is there data supporting reduced cGAMP synthesis or inhibition of SIN on direct cGAS activity?

Response 20:

Thank you very much for your question. Unfortunately, we have not yet conducted experiments to reduce the effect of cGAMP synthesis or inhibit SIN on the direct activity of cGAS, so we cannot provide relevant data support at this time.

In the planning of this study, the initial focus was on the potential mechanisms of SIN in treating DN, and to verify the effects of SIN on physiological indicators, kidney pathology, and inflammatory factors (IL-1β, IL-6, TNF-α) as well as the expression of downstream proteins in the cGAS/STING pathway using in vivo experiments. Given that inflammation is a key factor in renal damage in DN, and the cGAS/STING pathway is closely related to inflammation, this study prioritized clarifying the regulatory role of SIN in inflammatory injury and the cGAS/STING pathway, laying the foundation for subsequent research. Therefore, it did not involve studies on the direct activity of cGAS and cGAMP synthesis by SIN. However, we fully recognize the importance of investigating the relationship between cGAS activity and cGAMP synthesis and SIN.

Given this, we plan to design and conduct relevant experiments in subsequent studies. Specifically, we will set up control groups and different concentration treatment groups of SIN, using appropriate detection methods (such as detecting cGAMP levels with the ELISA method, and assessing cGAS activity through in vitro enzyme assays) to investigate the direct impact of SIN on cGAS activity and its role in cGAMP synthesis mediated by cGAS.

Thank you again for your suggestions, which have pointed out the direction for our follow-up research. We will try our best to improve the content and quality of the research.

Comments 21:Is there long-term implication in a chronic disease model of SIN therapy?

Response 21:

Thank you for your question! SIN is widely used in China, particularly in the treatment of chronic diseases such as rheumatoid arthritis, kidney-related conditions, tumors, and psychoneurological disorders, although its mechanisms of action remain not fully understood. According to reports, adverse reactions caused by SIN in treating chronic diseases are often associated with mast cell activation. This study focuses on the renal protective effects of SIN regulating cGAS-STING in db/db mice. However, there is no evidence linking cGAS-STING to mast cells. In future research, we will delve deeper into the issues you raised.

Comments 22:Are species differences in activation or metabolism of STING occurring which limit translational relevance?

Response 22:

Thank you for pointing out the issue! The species differences in STING activation or metabolism to some extent limit its translational relevance. There are variations in the sequence and structure of STING proteins across different species, which directly affect their activation mechanisms and efficiency. For example, there is about a 40% difference in amino acid sequences between mouse and human STING, making mouse STING activation dependent on cGAMP concentration, while human STING has lower sensitivity to cGAMP. In terms of metabolism, cells from different species differ in the utilization of metabolic substrates, which in turn affects the metabolic regulation of STING. For instance, varying expression levels and activities of metabolic enzymes among different species can lead to differences in the production and clearance of STING-related metabolites. However, due to the differences in regulatory mechanisms of metabolic signaling pathways across species, this can also inversely impact the metabolic regulation of STING. These differences result in varying activation efficiencies of STING in different species, leading to differences in its ability to activate downstream signaling pathways and gene expression, thereby limiting its regulatory role at the translational level. Addressing this issue falls beyond the scope of our current study. We will delve deeper into this topic in future research.

Comments 23:If mouse and human STING differ, by what mechanism might relevance to human treatment be affected?

Response 23:

Thank you for your question! As mentioned above, the differences between STING in mice and humans indeed affect human therapeutic relevance. The mechanism of STING-related treatments in humans should be discussed from the perspective of species-specific differences in STING between animals and humans. For example, compared to mice, humans have lower sensitivity to STING and require higher levels of cGAMP to activate it. Additionally, different cell types respond to STING to varying degrees; for instance, STING activation in mouse dendritic cells leads to a large secretion of cytokines, whereas human dendritic cells secrete less. Therefore, the mechanisms of STING-related treatments are more related to metabolism and changes in the microenvironment within the body. Furthermore, cell type should also be considered in clinical treatment. Currently, there is still a lack of research on STING, and your question has pointed us in the right direction for future studies.

Comments 24:Is there proof for immunosuppression that follows STING inhibition?

Response 24:

Thank you for your question! At this stage, research on the secondary effects of STING inhibition is still insufficient. However, existing clinical and experimental evidence suggests that STING inhibition may indeed lead to immune suppression. For example, in the tumor microenvironment, activation of the STING pathway can promote the production of interferons and other related cytokines, inducing an anti-tumor immune response. Moreover, STING is associated with various immune cells; inhibiting STING may affect the normal function of these immune cells, thereby inducing immune suppression. This study focuses on the impact of cGAS-STING on DN and the renal protective effect of SIN in db/db mice. Our results suggest that cGAS-STING in db/bd

In mice, overactivation was observed, while SIN inhibited cGAS-STING to relieve kidney injury without immunosuppressive effects. In subsequent studies, we will further investigate the role of STING in DN.

Comments 25:How mechanistically and functionally is SIN different from known cGAS-STING inhibitors?

Response 25:

Thank you for your question! The current mechanisms and functions of cGAS-STING inhibitors mainly include targeting the active site of cGAS / STING ligand binding domain, interfering with the interaction between cGAS and double-stranded DNA, blocking cGAS dimerization, and inhibiting post-translational modifications of STING. In this study, although we observed the inhibitory effect of SIN on cGAS-STING, the specific mechanism has not been fully elucidated. We will continue to explore further in subsequent research.

Comments 26:Is the reduction in NF-κB activity cause or effect of modulation in upstream pathways?

Response 26:

Thank you for your question! The cGAS-STING and NF-κB pathways have a bidirectional regulatory mechanism. For example, after the cGAS-STING pathway is activated, the STING protein recruits and activates the kinases TBK1 and IKKε, promoting the phosphorylation of IRF3 to release IFN-γ into the nucleus. Similarly, in sepsis models, the activation of NF-κB has been observed to promote the activation of STING. In this study, the activation of cGAS-STING and NF-κB was evident in db/db mice with active immune inflammation. Our research focus is on the effects of cGAS-STING in db/db mice and the impact of SIN on the expression of cGAS-STING. Your question points out the direction for our future research, and we will delve deeper into this issue.

Round 2

Reviewer 2 Report

Comments and Suggestions for Authors

The paper can be accepted in its present form.